# The 10-Item Short Form of the German Experiences in Close Relationships Scale (ECR-G-10)—Model Fit, Reliability, and Validity

**DOI:** 10.3390/bs13110935

**Published:** 2023-11-16

**Authors:** Eva Neumann, Elke Rohmann, Heribert Sattel

**Affiliations:** 1Department of Psychosomatic Medicine and Psychotherapy, LVR University Hospital Düsseldorf, Heinrich Heine University, 40629 Düsseldorf, Germany; 2Department of Social Psychology, Faculty of Psychology, Ruhr University, 447801 Bochum, Germany; elke.rohmann@rub.de; 3Department of Psychosomatic Medicine and Psychotherapy, Klinikum Rechts der Isar, Technical University, 81675 Munich, Germany; h.sattel@tum.de

**Keywords:** adult attachment, avoidance, anxiety, experiences in close relationships scale, ant colony optimization

## Abstract

The aim of the present work was the development and validation of a short form of the Experiences in Close Relationship Scale (ECR) in German. Three studies were conducted. In study 1, the best items for the short form were selected from the item pool of the original version based on ant colony optimization (ACO), a recently developed probabilistic approach. Data from three samples collected at a university, an online portal, and a psychosomatic clinic with a total of 1470 participants were analyzed. A 10-item solution resulted, measuring avoidance and anxiety with five items each. This solution showed a good model fit and acceptable reliability in all three samples. The two new short scales were independent of each other. In study 2, the 10-item solution was validated by correlating the new short scales with external criteria. Data from previous studies that included student, community, and clinical samples were reanalyzed. Both short scales showed expected correlations with measures of romantic relationships, personality, psychopathology, and childhood trauma, indicating convergent and discriminant validity. The significant correlations were moderate to strong. In study 3, the selected ten items alone and several content-related scales were presented online to 277 participants, most of them students. The good results in terms of model fit, reliability, and validity observed in studies 1 and 2 could be replicated here. The new short form, called ECR-G-10, allows the measurement of attachment avoidance and anxiety in an economic way in research and clinical practice.

## 1. Introduction

In the 1980s, Hazan and Shaver [1] established a new tradition in attachment theory and research by presenting the first measure of attachment in adult romantic relationships. Based on the three-category model by Ainsworth et al. [2], they formulated an item related to romantic relationships in adulthood for each of the attachment styles: secure, anxious-ambivalent, and avoidant. Respondents are asked to select the item that best describes their experiences and behavior. The new measure made it possible to extend attachment research, which previously focused on children’s relationships with their caregivers, to close relationships between adults. The pioneering work of Hazan and Shaver inspired researchers worldwide and led to numerous studies of adult attachment [3].

Another milestone was the introduction of the Experiences in Close Relationships Scale (ECR) by Brennan, Clark, and Shaver [4]. It quickly became clear that the 3-item measure by Hazan and Shaver has methodological limitations due to its brevity. In the years following the publication of this measure, several multi-item questionnaires for the measurement of adult attachment were developed. Brennan et al. [4] factor-analyzed the items of all questionnaires available at the time, a total of 323 items. They found a two-dimensional structure. The first dimension, avoidance, stands for discomfort in situations of intimacy and an emphasis on self-reliance in romantic relationships. The second dimension, fear, represents intense worry about the relationship, including a fear of abandonment and a strong need for attention and care from one’s partner. Brennan et al. [4] selected the best 18 items for each of the two dimensions, resulting in the final 36-item version of the ECR.

The ECR has become a widely used instrument and has been translated into 17 languages. It is often considered the gold standard for measuring adult attachment. The psychometric properties of this instrument have repeatedly proven to be excellent [3]. Most studies found alpha coefficients of around 0.90 for both scales, indicating high reliability. The correlation between the two scales is usually close to zero, showing that the scales are independent of each other.

Furthermore, the ECR has proven to be valid in hundreds of studies [3]. The two scales correlated as expected with other measures of romantic relationships. A variety of studies with different research designs (e.g., cross-sectional and longitudinal studies, diary studies, and experiments) found that low scores for avoidance and anxiety were associated with a positive view of the relationship and higher satisfaction with the relationship and sexuality [5]. Furthermore, significant correlations with measures of personality and psychopathology could be shown. For example, the ECR scales were found to correlate with the Big Five of personality, with the strongest correlation observed between anxiety and neuroticism [6]. Such associations have also been demonstrated for pathological personality traits in psychiatric patients. Here, the highest correlations were found between anxiety and negative affectivity and between avoidance and detachment [7]. Finally, the ECR scales turned out to correlate with measures of mental disorders. Many of the studies in this area have focused on depression, anxiety, or borderline personality disorder, with all three of these mental disorders showing positive associations with attachment avoidance and anxiety [3,8].

Fraley et al. [9] re-analyzed the 323 items from the study by Brennan et al. based on item-response theory. Their resulting selection of 36 items differed in part from that of Brennan et al. [4]. The new instrument, called the Experiences in Close Relationships Scale-Revised (ECR-R), proved to be equivalent to the ECR in terms of reliability and validity, but its two subscales showed a higher intercorrelation [10]. This is likely why the ECR-R has not replaced the ECR in the years since its release; both instruments continue to be used in adult attachment studies.

The ECR-R was also translated into German [11]. In addition to the long form, two short forms of the German ECR-R are available, a 12- and an 8-item version [12,13]. The two-dimensional structure of the long form could be replicated for the short forms in the two studies conducted to develop them. However, this result could not be confirmed recently in another study, where both the 12-item and 8-item versions showed poor model fit [14]. Therefore, the structural validity of the two German ECR-R short forms could not be consistently demonstrated, leaving open the question of whether these versions adequately measure the two attachment dimensions.

Our research group developed the German version of the ECR [15]. The aim of the present study was to establish a short form of this instrument, an endeavor that seemed worthwhile for several reasons. Although the ECR is not overly long, there are study designs that require an even shorter version. This is especially the case when several other instruments are used in addition to the ECR. Furthermore, sometimes participants cannot be expected to complete a lengthy survey, for example, if they do not receive payment or course credit or if they have limited resilience due to mental illness. We did not consider using the short forms of the ECR-R for such studies but decided to develop a short form of the ECR. This is based primarily on the view that the ECR and ECR-R should be regarded as different instruments because they have different items, especially in the German versions due to divergent translations of some items. It is therefore likely that the constructs measured by the ECR and ECR-R are not exactly the same. Furthermore, another factor was that the test quality of the ECR-R short forms could not be consistently proven.

For the original American version of the ECR, 12-item versions have already been developed by Wei et al. [16] and Lafontaine et al. [17]. However, it was not promising to simply adopt one of these item selections for the German short form. Both selections include items that only showed average values for item characteristics (factor loadings, corrected item-total correlations) in our studies, in contrast to other items that performed better in this regard. Consistent with these concerns, the use of the selection by Wei et al. [16] in a sample from the normal population showed that this item selection in the German version lacks reliability and factorial validity [18]. Therefore, we decided to develop a new short version with the best items from the German long version of the ECR.

Another aim of this study was to overcome a shortcoming of the long version that emerged when we used it in our studies. As with the original American version, the term for the partner varies across the 36 items [3] (p. 85). The items refer to the romantic partner with the terms “my partner”, “a partner”, and “partners”. In our studies, “my partner” did not cause any problems, while “a partner” and “partners” sometimes led to confusion and problems of understanding; some participants were not sure which person to refer to when rating items with these terms. In order to eliminate these inconsistencies, we wanted to achieve an item selection in which the partner is consistently referred to as “my partner”.

We conducted three studies to develop and validate the short version of the German ECR, called ECR-G-10 in the following. The aim of the first study was to select the best items of the ECR long form for the short form. For this purpose, we analyzed data from studies in which we used the ECR long form, called ECR-G-36 in the following. The second study is based on re-analyses of data from former studies that also used the ECR-G-36. For the validation of the ECR short form, we examined whether the correlations with external criteria obtained with the short form correspond to those obtained with the long form. The aim of the third study was to prove that the ECR-G-10 is reliable and valid when its ten items are presented alone. Therefore, this instrument was used together with other content-related scales and analyzed with regard to its test quality.

## 2. Study 1

To select the best items for the short version of the ECR, we analyzed data from three samples. The data from the first and the third samples were collected in previous studies and re-analyzed for the present study, and the data from the second sample were newly collected. Data collection took place at a university, via an internet platform, and in a psychosomatic clinic. Since the first sample had the highest number of participants, these data were used for the initial selection of items for the short version. The data from the second and the third samples served as the replication of this selection.

### 2.1. Methods

#### 2.1.1. Participants and Settings

**Sample 1**. Sample 1 was recruited at the Faculty of Psychology of the Ruhr University Bochum for a study on romantic relationships [19]. Participants had to fulfill the criterion of living in a heterosexual relationship. They were contacted by email, on internet platforms, and in the media. Data were collected online via Unipark (www.unipark.de, accessed on 4 October 2020). A total of 788 persons took part in the study, 552 women (70%) and 236 men (30%) with a mean age of 28.29 years (*SD* = 8.58, range 18–60 years). 380 (48%) were students, 312 (40%) were employed, 38 (5%) were in training, 28 (4%) were homemakers, 28 (4%) were unemployed, and 2 (<1%) did not report their professional status.

**Sample 2**. Sample 2 was recruited for the present study via the internet platform prolific (www.prolific.co, accessed on 4 October 2020). A total of 220 participants took part in the study, 108 women (49.10%), 111 men (50.5%), and 1 unspecified (0.5%) with a mean age of 31.05 years (*SD* = 8.97). All participants had a romantic partner, 166 (75.5%) were dating, and 54 (24.5%) were married. A total of 78 (35.5%) were students, 4 (1.8%) apprentices, 115 (52.3%) employed, 11 (5.0%) unemployed, 8 (3.6%) homemakers, and 4 (1.8%) other.

**Sample 3**. Sample 3 included the participants of several clinical studies on associations between close relationships and psychopathology conducted in the Department of Psychosomatic Medicine and Psychotherapy of the LVR University Hospital Düsseldorf (e.g., published in [20,21]). Data were collected via paper-and-pencil surveys.

The sociodemographic and clinical data of the participating patients were quite similar across the studies. In all studies, about two-thirds of the patients were female and one-third were male, the mean age was between 38 and 48 years, and all patients were diagnosed with a mental disorder of moderate severity. It was therefore considered appropriate to combine the data from the various studies into one data set.

A total of 462 in- and outpatients from the hospital participated in the studies, 306 (66%) women and 156 men (34%) with a mean age of 43.59 years (*SD* = 12.51, range 18–77 years). The most common primary diagnosis was a somatoform disorder (*n* = 277, 60%), followed by a depressive disorder (*n* = 107, 23%); other diagnoses were less common (anxiety, adjustment, eating, or other disorders: *n* = 78, 17%).

#### 2.1.2. Measures

**Experiences in Close Relationships Scale—German version (ECR-G-36)**. In all three samples, the ECR-G-36 was used. The items are assessed on a 7-point Likert scale from 1 (*disagree strongly*) to 7 (*agree strongly*). The mean of the 18-item ratings per scale are the avoidance and anxiety scores.

#### 2.1.3. Content-Based Pre-Selection

As argued above, 11 items of the ECR-G-36 had the problematic wording “partners” and “a partner” (1, 3, 9, 12, 13, 21, 23, 29, 31, 32, 34). (It should be noted that “romantic partner(s)” was also translated to “my partner” in the German version because there is no term in German that exactly corresponds with this American term.) In the following analyses, an item selection was considered that no longer contained these 11 items, but consisted of the 25 items without the problematic formulation mentioned above.

#### 2.1.4. Statistical Analysis

The item selection for each subscale was based on consecutive analyses, which have been tailored for this purpose and were recently compared by Schroeders et al. [22]. Both subscales were evaluated separately, as their independence was theoretically suggested and proof of that assumption was integrated into the analysis procedure.

Firstly, a stepwise confirmatory factor analysis (SCOFA) was conducted on both scales. The SCOFA algorithm iteratively removed the item with the lowest factor loading from the item pool and indicated an optimal length for a short form according to the estimated information criteria.

Secondly, we conducted an ant colony optimization (ACO) following the approach of Schroeders et al. [22] and Volz et al. [23]. The generation of short versions of questionnaires can be understood as an optimization of the item selection procedure. The ACO provides a probabilistic algorithm for this purpose, which was reported and mathematically described by Feynman [24], from which its denotation is derived. This class of algorithms is based on an iterative and combinatorial process that imitates ants searching for food. For this purpose, ants aim to find the shortest path marked by a chemical trace of pheromones. Because these pheromones are constantly evaporating, shorter paths have a higher concentration of them and are therefore more attractive to following ants. The shortest path in the context of determining an item selection can be operationalized as an optimal quality criterion that must be determined mathematically.

The ACO starts with the selection of a random sample based on all suitable items. Then, predefined quality criteria of interest (e.g., model fit and reliability) of this sample are determined, and subsequently, a large number of different randomly selected samples are compared. The algorithm leaves the best at the top to identify an optimized solution. Considering the fact that this probabilistic approach cannot determine a theoretically existing optimal solution in every case, a series of repetitions of this procedure is recommended [23].

The calculation of the optimization criterion for our approach was based on four criteria. Firstly, the model fit was defined by a composite and equivalent score out of the comparative fit index (CFI) and the root mean square error of approximation (RMSEA), with CFI values beyond 0.95 and RMSEA values lower than 0.05 as indicators of appropriate model fit [25]. Secondly, the measurement precision of the scale was estimated, defined by McDonald’s omega, an index that determines the saturation of a factor in a unidimensional model. We assumed that values greater than 0.70 indicate acceptable reliability. Thirdly, the correlation of the actual short form with the long form of the scale was calculated, as a measure of criterion validity. Here, a correlation of 0.80 was chosen as the criterion. Finally, as the independence of both subscales was theoretically postulated, we included this parameter in our estimation of the overall optimization criterion and set a correlation of less than 0.10 as desirable. Each of these four parameters was logit-transformed, resulting in a value range between 0 and 1. An additional weight term was introduced, so that small undesired differences close to the predefined cutoff led to a proportionally more pronounced decrease in the parameters’ value (for details see [22,23]). The overall optimization criterion was built as the sum of all four logit-transformed parameters. While this procedure can identify solutions that come close to or meet all of the above criteria, it does not necessarily identify the potential best solution. Thus, ACOs were run with 80 ants per iteration, an evaporation rate of 0.9, and 50 iterations without improvement of the overall criterion as a stop condition. Each ACO was run 15 times with random initial seeds, and the best solution in sample 1 was selected. It was then checked whether the individual criteria were also met in the other two samples.

Finally, we conducted a two-factorial confirmatory factor analysis with the selected items to determine the factorial model. For reporting issues and the sake of comparability, Cronbach’s alpha was additionally determined.

The analyses were conducted using R x64 version 4.0.4 including the packages lavaan and psychtools, as well as IBM SPSS 27.

### 2.2. Results

In the SCOFAs, all 36 items of the ECR were considered. The SCOFA for the subscale “anxiety” resulted in an elimination of all items with an unspecific reference to the partner in the first steps and identified a solution with satisfying model fit, reliability, and criterion validity, but not independence. The same procedure did not result in such a model for the subscale “avoidance”. Both SCOFAs identified a 5-item solution as optimal, considering the information criteria of each stepwise generated model.

Then, an ACO was carried out for each subscale based on the pool of the 25 pre-selected items with the data from sample 1. For avoidance, one solution turned out to show the highest value of the overall optimization criterion (considering model fit, reliability, validity, and independence) in sample 1, the solution consisting of items 7, 15, 25, 27, and 35. These characteristics could be confirmed in both replication samples, except for independency within clinical sample 3, where a correlation of 0.18 with the full scale for anxiety was observed.

The same procedure for the subscale “anxiety” resulted in several solutions that met the prespecified requirements within the sample of study 1. The overall optimization criterion values were almost equal in these solutions. Therefore, additional content-related aspects were considered in further selection. Solutions that had one or more of the following weaknesses were not considered further:Selections containing both items 2 and 22 were rejected. These items have the same wording in the German version except for one word. A short version with only five items per subscale should not include two items with almost the same wording.Solutions where 4 of the 5 items started with the wording “I worry” were rejected. These solutions therefore have repetitive formulations, making the selected items for the anxiety scale very similar.Solutions containing item 28 were rejected. This item relates to periods of life when the subjects were single and is therefore difficult for people in a long-term relationship to answer.

After this step, two solutions remained that differed in only one item. These solutions both contained the four items 18, 20, 24, and 30. The fifth item was item 2 in one solution and item 8 in the other. Both items refer to the fear of being abandoned. Item 8 was given preference because this item specifically addresses romantic relationships, while the wording of item 2 is more general and does not refer to a specific kind of relationship. The final solution selected for the anxiety short scale was therefore 8, 18, 20, 24, and 30. The characteristics (considering model fit, reliability, validity, and independence) of the resulting model were confirmed in both replication samples, again except for independence within samples 2 and 3, where correlations of 0.18 and 0.15, respectively, with the full scale for avoidance were observed.

The top rows of Table 1 show the model fit of the two selected solutions. In all three samples of study 1, RMSEA was between 0.01 and 0.07, and CFI was larger than 0.99, thus reaching adequate to good parameters. The reliability, as indicated by McDonald’s omega and Cronbach’s alpha, was acceptable.

The factorial model of the selected solution revealed sufficiently high loadings of all items on the respective factor to which they are assigned. Figure 1 shows the latent measurement model. Depicted are standardized factor loadings resulting from all three samples. The loadings were between 0.40 and 0.94 (all loadings *p* < 0.001). Both factors proved to be widely independent (standardized covariances of 0.01 to 0.14 for all samples).

The correlations between the two new short scales were low and did not reach statistical significance (correlation avoidance—anxiety in sample 1: *r* = 0.06, *p* = 0.10, in sample 2: *r* = 0.08, *p* = 0.27, in sample 3: *r* = 0.03, *p* = 0.60). These correlations, which did not reach the threshold for a small effect size [26], indicated that the two new short scales were independent of each other.

## 3. Study 2

Study 2 served as the validation of the ECR-G-10. For this purpose, we re-analyzed data from former studies that used the ECR-G-36 [21,27,28,29]. In these studies, the ECR was correlated not only with alternative scales for the measurement of romantic relationships but also with scales measuring variables of personality, psychopathology, and childhood trauma. Rohmann et al. [27] focused on romantic relationships, Makuch and Neumann [28] on psychopathology and the Big Five of personality, Neumann [21] on personality disorders, and Neumann [29] on childhood trauma. We computed correlations of the ECR long and short form with these variables. The hypothesis was that the correlations of the two forms are equal in direction (positive or negative) and statistical significance, showing that the short form is equivalent to the long form.

### 3.1. Methods

#### 3.1.1. Participants and Settings

The four studies considered in this analysis included samples from student, normal, and clinical populations.

In the study by Rohmann et al. [27], 92 participants were recruited at the Ruhr University Bochum, most of them students. The mean age was 24.03 years (*SD* = 5.40), 61 (66.3%) were women, and 31 (33.7%) were men. Participation in the study required that the participants were in a romantic relationship. The average relationship length was 37.08 months (*SD* = 40.87).

In the study of Makuch and Neumann [28], a community sample was recruited in a dental practice in Düsseldorf. A total of 110 persons took part, 69 women (62.7%) and 41 men (37.3%). The mean age was 38.61 years (*SD* = 14.75). Thirteen participants (11.8%) were singles, 44 (40.0%) had a dating partner, 40 (36.4%) were married, and 13 (11.8%) were divorced or widowed. In terms of educational attainment, 33 (30%) had a secondary school diploma, 30 (27.3%) had a university entrance qualification, and 44 (40.0%) had a university degree.

The sample of the study of Neumann [21] consists of in- and outpatients of the Department of Psychosomatic Medicine and Psychotherapy of the LVR University Hospital Düsseldorf. A total of 110 patients (65 women (59.1%) and 45 men (40.09%)) with a mean age of 40.85 years (*SD* = 12.99) took part in the study. Thirty-five (31.8%) were singles, 23 (20.9%) had a dating partner, 38 (34.5%) were married, and 14 (12.7%) were divorced or widowed. Regarding the level of education, 35 (31.8%) had a secondary school diploma, 28 (25.5%) had a university entrance qualification, and 46 (41.8%) had a university degree. A depressive disorder was the most frequent main diagnosis (80, 72.7%).

In the study of Neumann [29], two samples were compared, 80 patients diagnosed with a depressive disorder from the LVR University Hospital Düsseldorf and the 110 patients of the dental practice described above. Together, data from 190 participants were analyzed, 120 women (63.2%) and 70 men (36.8%) with a mean age of 39.36 years (*SD* = 13.94).

#### 3.1.2. Measures

**Experiences in Close Relationships Scale, German version (ECR-G-36)**. In all four studies, the ECR-G-36 was used for the measurement of adult attachment.

**Relationship Assessment Scale (RAS)**. The RAS is a one-dimensional scale for the measurement of satisfaction with romantic relationships [30] (German version [31]). The seven items of this instrument are answered on a 7-point Likert scale from 1 (*not at all satisfied*) to 7 (*very satisfied*).

**Marburg Attitude Scales towards Love Styles (MEIL)**. The six love styles (erotic, game-playing, friendship, pragmatic, possessive, and altruistic love) proposed by Lee [32] stand for different attitudes towards the romantic partner. The MEIL, a questionnaire that is available in a 60-item version and a 30-item short version, allows these styles to be measured with a response format ranging from 1 (*not at all true*) to 9 (*very true*) [33].

**Symptom Checklist (SCL)**. The 90 items of the SCL describe mental and somatic symptoms [34] (German version [35]). The level of stress caused by these symptoms is rated on a 5-point Likert scale ranging from 0 (*not at all*) to 4 (*extremely*). The total score Global Severity Index (GSI) indicates the general level of stress in psychological, social, and physical terms.

**Hospital Anxiety and Depression Scale (HADS)**. This instrument was developed for patients with somatic or mental disorders and serves as the measurement of anxiety and depression [36] (German version [37]). In order to distinguish mental from physical complaints, the items relate specifically to psychological symptoms of depression and anxiety and do not ask for somatic symptoms. The 14 items of this instrument have an answer format from 0 to 3. The answers are worded differently and refer to the frequency or the severity of the respective symptom or to the extent of change in the symptom.

**Patient Health Questionnaire Somatic Symptom Severity Scale (PHQ-15)**. The PHQ-15, a module from the Patient Health Questionnaire, allows the rating of the severity of somatic symptoms [38] (German version [39]). The 15 items describe somatic symptoms and are rated on a 3-point Likert scale from 0 (*not bothering at all*) to 2 (*bothering a lot*).

**NEO-Five-Factor Inventory (NEO-FFI-30)**. The NEO-FFI serves as the measurement of the Big Five of personality—neuroticism, extraversion, openness, agreeableness, and conscientiousness [40] (German 30-item short version [41]). Each of the five personality dimensions is measured with six items that are answered on a 5-point Likert scale from 1 (*disagree strongly*) to 5 (*agree strongly*).

**Assessment of DSM-IV Personality Disorders (ADP-IV)**. The ADP-IV allows dimensional ratings of the ten personality disorders of the DSM-IV, which are also listed in the DSM-5 [42] (German version [43]). Each diagnostic criterion of the personality disorders is described in an item. The instrument includes a total of 94 items that are responded to on a 7-point Likert scale from 1 (*totally disagree*) to 7 (*fully agree*).

**Childhood Trauma Questionnaire (CTQ)**. The CTQ serves as the retrospective assessment of traumatic interpersonal experiences in childhood [44] (German version [45]). The 25 items are rated on a 5-point Likert scale from 1 (*never true*) to 5 (*very often true*). The CTQ results in scores for five subscales (emotional abuse, physical abuse, sexual abuse, emotional neglect, physical neglect).

### 3.2. Results

The correlations of the original long scales and the new short scales of the German ECR with the external variables are shown in Table 2. The results were very similar for the long and short form. In almost all cases, the correlations with external criteria had the same direction; i.e., they were both positive or both negative. With only a few exceptions, there was also concordance regarding the significance or non-significance of the correlations.

The correlation coefficients of the short scales were lower than those of the long scales, but the difference was small in most cases and did not have an impact on the main result, i.e., whether the correlation was positive or negative and significant or not significant.

The two short scales correlated differently with the external variables in many cases. Avoidance showed the highest significant correlations with relationship satisfaction, erotic love, and conscientiousness (negative) and with schizoid personality disorder and emotional abuse and neglect (positive). The effect sizes were moderate. Anxiety showed the highest significant correlations with possessive love, stress, general anxiety, somatization, neuroticism, and paranoid, histrionic, narcissistic, dependent, and compulsive personality disorder (all positive). The effect sizes were moderate to strong. The anxiety short scale therefore showed higher and more statistically significant correlations than the avoidance short scale, a pattern of results that was also found in the long scales.

In some cases, both ECR short scales were correlated with an external variable. This applied to depression, extraversion, schizotypal, antisocial, borderline, and avoidant personality disorder, and emotional abuse. These correlations were all positive except the correlation with extraversion.

## 4. Study 3

The aim of study 3 was to investigate whether the ECR-G-10 proves reliable and valid when its ten items are presented alone. For this purpose, new data were gathered in a university setting. Participants completed the ECR-G-10 and other self-report questionnaires measuring satisfaction with the romantic relationship, sexuality, and life, as well as self-esteem. Exploratory and confirmatory factor analyses were conducted for the short version. The other scales used in this study served as external criteria for the validity of the ECR-G-10.

### 4.1. Methods

#### 4.1.1. Participants and Settings

Participants were recruited via postings on social media platforms, by mail, and via bulletin boards at the Faculty of Psychology of the Ruhr University Bochum. They should be 18–35 years old and have been in a romantic relationship for at least three months. Psychology students received course credit for their participation. The questionnaires were filled out online via Qualtrics.

A total of 277 participants took part in the study, 176 women (63.5%) and 101 men (36.5%) with a mean age of 23.71 years (*SD* = 3.99). About two-thirds of the sample were students, 171 (61.7%), of whom 113 studied psychology and 58 other subjects. Of the non-student participants, 75 (27.1%) were employed, 17 (6.1%) were in training, and 14 (5.1%) reported another occupation or did not provide information on this item. Regarding sexual orientation, 231 (83.4%) indicated a heterosexual, 7 (2.5%) a homosexual, and 38 (14.1%) a bisexual orientation. The participants’ romantic relationships had a mean duration of 34.32 (SD 29.59) months.

#### 4.1.2. Measures

**Experiences in Close Relationships Scale, German 10-item version (ECR-G-10)**. The ECR-G-10 was used for the measurement of attachment avoidance and anxiety (Table 3). The order of the items in the new short form followed these considerations: As in the long form, the items for avoidance and anxiety were presented alternately. The measure should start and end with items with a positive meaning. However, this could only be carried out for the first item, belonging to the avoidance scale, because all items of the anxiety scale are negatively worded. Therefore, the item of the anxiety scale that was rated the least negative in study 1 was presented as the last item.

Congruent with the long form, the items of the ECR-G-10 are answered on 7-point Likert scales ranging from 1 (*disagree strongly*) to 7 (*agree strongly*). Items 1, 3, 7, and 9 are to be recoded. The score for avoidance is the mean of the items with odd numbers and the score for anxiety is the mean of the items with even numbers. At least four item answers per scale are required for the scoring. In studies that include participants who do not currently have a romantic partner, it is recommended that the instruction be accompanied by a note that these participants should refer to their most recent romantic relationship when responding.

**Relationship Assessment Scale (RAS)**. The RAS already described in the methods section of study 2 was used to measure satisfaction with the romantic relationship [30] (German version [31]).

**Satisfaction with Sexuality (SWS)**. Satisfaction with sexuality in the romantic relationship was assessed with the following item formulated by Rohmann and Bierhoff [46]: “How satisfied are you with your sexual relationship with your partner?” The response format is the same as for the RAS.

**Satisfaction with Life Scale (SWLS)**. General life satisfaction was measured with the SWLS [47] (German version [48]). The five items are answered on a 7-point Likert scale from 1 (*strongly disagree*) to 7 (*strongly agree*).

**Rosenberg Self-Esteem Scale (RSES)**. The RSES serves as the measurement of self-esteem [49] (German version [50]). The ten items with statements about the self are rated on a 4-point Likert scale from 1 (*strongly disagree*) to 4 (*strongly agree*).

#### 4.1.3. Statistical Analyses

Means and total-item correlations were calculated for the ten items of the new ECR short form. An exploratory factor analysis was conducted to test whether the two-factor structure of the long form could be replicated for the short form. Confirmatory factor analysis was performed to check model fit. The model fit indices were the root mean square error of approximation (RMSEA) and the comparative fit index (CFI). McDonald’s omega and Cronbach’s alpha again served as reliability indices. To test whether the two subscales of the ECR-G-10 were independent of each other, their intercorrelation was determined. Finally, the two subscales were correlated with the other scales used in this study. Pearson correlation coefficients were determined.

### 4.2. Results

Table 3 shows the results obtained for the ECR-G-10 items. As can be seen from the mean values, agreement was higher for the anxiety items than for the avoidance items. Item–scale correlations indicated very good discrimination (*r_it_* > 0.40) for nine items and good discrimination (*r_it_* > 0.30) for one item.

The exploratory factor analysis yielded the following eigenvalues:

2.79, 2.43, 0.90, 0.74, 0.70, 0.68, 0.53, 0.50, 0.40, 0.34

The eigenvalues decreased significantly after the second factor, confirming the expected two-factor structure. The explained variance of the first two factors was 52.2%.

The ten items loaded as expected on the two factors, with the items with odd numbers loading on the factor representing avoidance and the items with even numbers loading on the factor representing anxiety (Table 3). These loadings were very high, falling within a range of 0.51–0.81. The loadings on the factor to which the items do not belong were always low and mostly close to 0.

The results obtained for the two ECR-G-10 subscales are shown in the bottom rows of Table 1. The confirmatory factor analysis yielded values for RMSEA and CFI that showed a relatively good model-data fit. McDonald’s omega and Cronbach’s alpha indicate that the reliability of both scales was acceptable.

The correlation between the avoidance and the anxiety scale was *r* = −0.03, *p* = 0.68, indicating that the two scales were independent.

Table 4 shows the correlations of the ECR short scales with the scales for the measurement of satisfaction and self-esteem. Avoidance was negatively correlated with self-esteem and all three measures of satisfaction. Anxiety showed negative correlations with relationship satisfaction and self-esteem. The correlations of the two short scales varied in magnitude. Attachment avoidance was more strongly correlated with the satisfaction scales, while attachment anxiety showed a stronger correlation with self-esteem.

## 5. Discussion

The aim of the present work was to develop a short version of the German Experiences in Close Relationships Scale. For this purpose, three studies were conducted to select items from the long version and to demonstrate the reliability and validity of the resulting item selection. The studies used data from previous studies as well as newly gathered data.

In the first study, an ant colony optimization was conducted for the selection of items from the long version, consecutively in three different samples with a total of 1470 participants. A 10-item solution resulted, which measures attachment avoidance and anxiety with five items each. This solution, called the ECR-G-10, showed a good model fit not only in the initial sample but also in both replication samples. The finding was confirmed in study 3, in which the ten selected items were used alone in a sample of 277 participants. A good model fit was also shown here. The ten items loaded as expected on the two factors representing attachment avoidance and anxiety, again in all three samples of study 1 and in study 3.

The reliability of the new short scales proved to be acceptable. Cronbach’s alpha ranged from 0.7 to 0.8 in the samples of studies 1 and 3. These alpha values are lower than those of the long scales of the ECR-G-36, which are usually above 0.9. However, these very high values for Cronbach’s alpha may indicate some redundancy. The ACO excludes items that are highly correlated with another item due to similar wording, thus reducing redundancy. This probably led to a slight decrease in the internal reliability of the two short scales compared to the long scales. In addition, a decrease was also to be expected simply because the ECR-G-10 has a significantly smaller number of items than the ECR-G-36.

As intended, the two short scales showed a low inter-correlation. The coefficients of the correlation between avoidance and anxiety did not reach the threshold for a small effect size and were even close to zero in studies 1 and 3. The new short scales were thus found to be independent of each other.

Finally, the correlations of the two short scales with scales measuring romantic relationships, personality, psychopathology, and childhood trauma demonstrated their convergent and discriminant validity. In studies 2 and 3, avoidance and anxiety correlated similarly with the external criteria in some cases and differently in others. The correlations with depression, some personality disorders, and emotional abuse in childhood were almost the same. Differences were found in that avoidance correlated more strongly with scales assessing romantic relationships and childhood neglect, and anxiety correlated more strongly with scales assessing personality and psychopathology. These findings suggest that avoidance is more likely to reflect interpersonal experiences and behaviors in close relationships, whereas anxiety is more related to intrapersonal processes related to mental health. In summary, the correlational analysis supported theoretical assumptions about the nature of the two attachment dimensions and indicated the validity of the ECR-G-10 subscales.

Overall, consistent results emerged across the three studies in terms of model fit, factor structure, and validity of the ECR-G-10. The studies all showed a high test quality of the new short form. This agreement indicates the reliability of the results. Moreover, the fact that the results were similar in samples with very different participants (students, professionals, or psychotherapy patients) shows that the ECR-G-10 is applicable in a wide range of populations.

In addition to the statistical features outlined above, the ECR-G-10 has some strengths in terms of content. A weakness of the long version, the vague reference to the partner in items with the wording “partners” and “a partner”, was remedied by not including these items in the short version. This ensured that the partner is consistently referred to as “my partner” in all items of the ECR-G-10 in which the partner is explicitly mentioned.

Furthermore, although the ECR-G-10 is quite short, the content of the items reflects the essential characteristics of attachment. One item of the avoidance scale relates to avoidance of intimacy (5), and the other items represent using the partner as a secure base (1, 3, 7, 9). One item of the anxiety scale reflects the fear of being abandoned (4), and the other items describe a desire for the partner to show more love and care (2, 6, 8, 10). Thus, despite its brevity, the ECR-G-10 covers the core features of attachment.

### Limitations

Some aspects of our work limit the strengths of these analyses. The applied methods for the item selection were probabilistic. Potentially, estimating all possible combinations could yield even better solutions. In addition, it could be discussed whether modified or different optimization criteria should be applied for certain scenarios.

Women were overrepresented among the participants, making up two-thirds of the samples in this study (except sample 2 of study 1). It is therefore possible that the male perspective on romantic attachment and related constructs has not been sufficiently reflected here.

Another limiting factor was that all data came from self-report measures. It would therefore be interesting to investigate whether the ECR-G-10 also proves to be valid when comparing this instrument with measures based on other methodologies, such as scales for expert ratings or biological parameters. Furthermore, the scales for the assessment of romantic relationships and sexuality may have been difficult to answer for participants who were in an open relationship and felt connected to more than one partner.

## 6. Conclusions

The short form of the German Experiences in Close Relationship Scale (ECR-G-10) presented in this work allows for an economic measurement of adult attachment. The two dimensions, avoidance and anxiety, are assessed with only five items each. To develop and validate the short form, three studies were conducted that included samples from student, community, and patient populations. The ECR-G-10 showed a good model fit and acceptable reliability. The factorial structure of the ECR long form could be replicated for the short form. The two scales of the ECR-G-10 correlated as expected with scales for the measurement of romantic relationships, personality, psychopathology, and childhood trauma, indicating convergent and discriminant validity. The ECR-G-10 can be used in research studies as well as in clinical practice.

## Figures and Tables

**Figure 1 behavsci-13-00935-f001:**
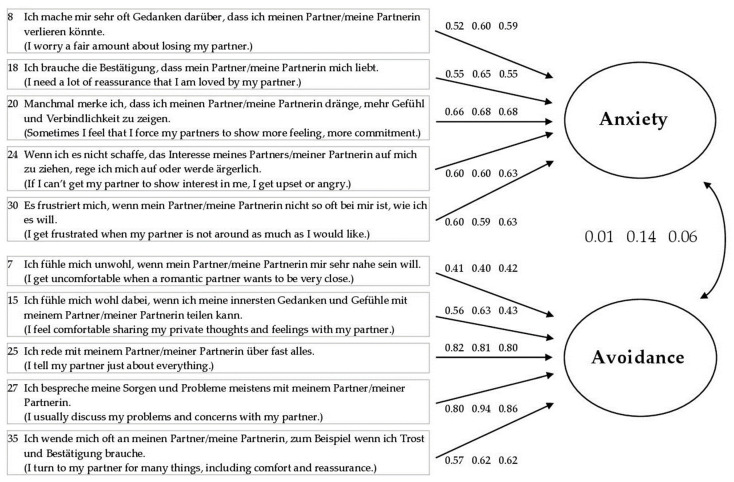
Standardized factor loadings of the ten items selected for the ECR-G-10.

**Table 1 behavsci-13-00935-t001:** Model fit, reliability, and correlation with the long scales of the ECR-G-10.

				Model Fit						Reliability		Correlation with Corresponding Long Scale	Correlation with Opposite Long Scale
Study	Sample	*n*		*df*	*Χ* ^2^	*p*	*RMSEA*	*90% CI*	*CFI*	*ω*	*α*	*r*	*r*
1	1	788	Avoid	5	5.88	0.32	0.02	0.00–0.05	0.99	0.80	0.77	0.93	0.07
			Anx	5	10.10	0.07	0.04	0.00–0.07	0.99	0.73	0.72	0.89	0.06
	2	220	Avoid	5	5.23	0.39	0.01	0.00–0.10	0.99	0.84	0.81	0.91	0.10
			Anx	5	7.57	0.18	0.05	0.00–0.11	0.99	0.76	0.76	0.91	0.18
	3	462	Avoid	5	10.90	0.05	0.05	0.00–0.09	0.99	0.78	0.75	0.88	0.18
			Anx	5	17.80	0.01	0.07	0.04–0.11	0.99	0.77	0.76	0.89	0.15
3		277	Avoid	5	16.59	0.01	0.09	0.05–0.14	0.97	0.78	0.78		
			Anx	5	10.29	0.07	0.06	0.00–0.12	0.98	0.73	0.73		

Note, *n* = number of participants, *df* = degrees of freedom, *Χ*^2^ = chi-square, *RMSEA* = root mean square error of approximation, *CI* = confidence interval, *CFI* = comparative fit index, *ω* = McDonald’s omega, *α* = Cronbach’s alpha, *r* = Pearson correlation coefficient, Avoid = avoidance, Anx = anxiety.

**Table 2 behavsci-13-00935-t002:** Correlations of the ECR-G long and short form with external criteria.

Data Source	*n*	Scale	Avoidance		Anxiety	
			ECR-G-36	ECR-G-10	ECR-G-36	ECR-G-10
Rohmann et al. [27]	92	Relationship satisfaction	−0.54 ***	−0.41 ***	−0.10	0.00
		Erotic love	−0.43 ***	−0.34 **	−0.05	0.06
		Game-playing love	0.10	0.02	0.07	0.07
		Friendship love	−0.11	−0.19	0.05	0.07
		Pragmatic love	−0.03	−0.05	0.07	0.07
		Possessive love	−0.25 ***	−0.25 *	0.56 ***	0.52 ***
		Altruistic love	−0.19	−0.13	0.22 *	0.23 *
Makuch and Neumann [28]	110	Global Severity Index	0.21 *	0.14	0.52 ***	0.42 ***
		Anxiety	0.26 ***	0.13	0.57 ***	0.47 ***
		Depression	0.38 ***	0.30 **	0.45 ***	0.37 ***
		Somatization	0.15	0.05	0.39 ***	0.33 ***
		Neuroticism	0.31 **	0.14	0.54 ***	0.43 ***
		Extraversion	−0.35 ***	−0.30 **	−0.29 **	−0.30 **
		Openness	−0.05	−0.13	−0.02	−0.04
		Agreeableness	−0.30 **	−0.13	−0.15	−0.15
		Conscientiousness	−0.34 ***	−0.30 **	−0.11	−0.07
Neumann [21]	110	Paranoid PD	0.27 *	0.18	0.55 ***	0.50 ***
		Schizoid PD	0.35 ***	0.38 ***	−0.02	−0.02
		Schizotypal PD	0.36 ***	0.33 **	0.32 **	0.29 **
		Antisocial PD	0.23 *	0.25 **	0.28 **	0.27 **
		Borderline PD	0.36 ***	0.33 ***	0.49 ***	0.45 ***
		Histrionic PD	0.21 *	0.18	0.48 ***	0.45 ***
		Narcissistic PD	0.19	0.19 *	0.43 ***	0.44 ***
		Avoidant PD	0.44 ***	0.35 ***	0.38 ***	0.37 ***
		Dependent PD	0.15	0.11	0.48 ***	0.43 ***
		Compulsive PD	0.15	0.09	0.36 ***	0.39 ***
Neumann [29]	190	Emotional abuse	0.33 ***	0.25 **	0.35 ***	0.27 ***
		Physical abuse	0.18 *	0.19 *	0.15 *	0.09
		Sexual abuse	0.14	0.18 *	0.07	0.04
		Emotional neglect	0.46 ***	0.40 ***	0.26 ***	0.17 *
		Physical neglect	0.36 ***	0.32 ***	0.13	0.04

Note, PD = Personality disorder, *n* = number of participants, * *p* < 0.05, ** *p* < 0.01, *** *p* < 0.001.

**Table 3 behavsci-13-00935-t003:** Descriptive data, reliability, and factor loadings of the ECR-G-10 items.

No.		*M*	*SD*	*r_it_*	*a* _1_	*a* _2_
1	Ich rede mit meinem Partner/meiner Partnerin über fast alles. (r)	1.78	1.01	0.66	**0.81**	0.00
2	Manchmal merke ich, dass ich meinen Partner/meine Partnerin dränge, mehr Gefühl und Verbindlichkeit zu zeigen.	3.55	1.91	0.51	0.04	**0.73**
3	Ich wende mich oft an meinen Partner/meine Partnerin, zum Beispiel, wenn ich Trost und Bestätigung brauche. (r)	2.11	1.28	0.58	**0.75**	−0.16
4	Ich mache mir sehr oft Gedanken darüber, dass ich meinen Partner/meine Partnerin verlieren könnte.	3.14	1.77	0.46	0.08	**0.67**
5	Ich fühle mich unwohl, wenn mein Partner/meine Partnerin mir sehr nahe sein will.	1.67	1.26	0.35	**0.51**	0.01
6	Wenn ich es nicht schaffe, das Interesse meines Partners/meiner Partnerin auf mich zu ziehen, rege ich mich auf oder werde ärgerlich.	2.90	1.60	0.53	0.07	**0.72**
7	Ich bespreche meine Sorgen und Probleme meistens mit meinem Partner/meiner Partnerin. (r)	2.04	1.25	0.62	**0.79**	0.04
8	Es frustriert mich, wenn mein Partner/meine Partnerin nicht so oft bei mir ist, wie ich es will.	4.25	1.77	0.48	−0.05	**0.67**
9	Ich fühle mich wohl dabei, wenn ich meine innersten Gedanken und Gefühle mit meinem Partner/meiner Partnerin teilen kann. (r)	1.96	1.30	0.60	**0.78**	0.07
10	Ich brauche die Bestätigung, dass mein Partner/meine Partnerin mich liebt.	5.32	1.58	0.49	−0.21	**0.69**

Note, (r) = recoded, *M* = mean, *SD* = standard deviation, *r_it_* = item-total correlation, *a*_1_ = loading on factor 1, *a*_2_ = loading on factor 2.

**Table 4 behavsci-13-00935-t004:** Correlations of the ECR-G-10 with external criteria.

	Avoidance	Anxiety
Relationship satisfaction	−0.48 ***	−0.18 **
Sexual satisfaction	−0.47 ***	−0.02
Satisfaction with life	−0.26 ***	−0.12
Self-esteem	−0.19 **	−0.30 ***

Note, ** *p* < 0.01, *** *p* < 0.001.

## Data Availability

The data presented in this study are available on request from the corresponding author. The data are not publicly available due to patient data protection regulations. The authors will also provide the ACO code written for this study upon request.

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
