# Peer review of "The 10-Item Short Form of the German Experiences in Close Relationships Scale (ECR-G-10)—Model Fit, Reliability, and Validity"

_behavsci, 2023, doi:10.3390/bs13110935_

Round 1

Reviewer 1 Report

Comments and Suggestions for Authors

The present paper aimed to develop and validate a short form of the Experiences in Close Relationship Scale (ECR) in German. In general, the paper's current version addresses the study's objectives. However, specific revisions are needed, in particular for the method session.

2.1. Method. Please provide a clear description of the participants according to Table 1. Moreover, the procedures used to select the items and if they are coherent with the original factors are not discussed clearly.

Table 1 shows four studies!

Table 2. Most of the descriptions are inconsistent with the results shown in Table 2. For example, also Agreeableness is negative.

It is also essential to clarify why most results are insignificant!

Author Response

Response to Reviewer 1

Thank you for reviewing our manuscript and the constructive feedback. Please find the detailed responses below. Revised and newly formulated passages are quoted here and shown in red.

The present paper aimed to develop and validate a short form of the Experiences in Close Relationship Scale (ECR) in German. In general, the paper's current version addresses the study's objectives. However, specific revisions are needed, in particular for the method session.

2.1. Method. Please provide a clear description of the participants according to Table 1. Moreover, the procedures used to select the items and if they are coherent with the original factors are not discussed clearly.

The samples are described in the text (the three samples of study 1 on pp. 3,4, lines 141-168, the sample of study 3 on p. 10, lines 411-424). To avoid redundancy, we have not provided the characteristics of the sample again in the table. Please let us know if you think that the descriptions in the text are not sufficient.

The passage in which the ACO, the procedure that was used for the item selection, was extended, including a new reference (pp. 4,5, lines 192-207). We hope that the rationale of this procedure now becomes clearer.

The ACO provides a probabilistic algorithm for this purpose, which was reported and mathematically described by Feynman [26], from which its denotation is derived. This class of algorithms is based on an iterative and combinatorial process that imitates ants searching for food. For this purpose, ants aim to find the shortest path marked by a chemical trace of pheromones. Because these pheromones are constantly evaporating, shorter paths have a higher concentration of them and are therefore more attractive to following ants. The shortest path in the context of determining an item selection can be operationalized as an optimal quality criterium that must be determined mathematically.

The ACO starts with the selection of a random sample based on all suitable items. Then, predefined quality criteria of interest (e.g., model fit and reliability) of this sample are determined, and subsequently a large number of different randomly selected samples are compared. The algorithm leaves the best at the top to identify an optimized solution. Considering the fact that this probabilistic approach cannot determine a theoretically existing optimal solution in every case, a series of repetitions of this procedure is recommended [25].

Table 1 shows four studies!

That is true. After some consideration, we decided to combine the data on model fit from studies 1 and 3 in one table, since (except for the correlations with the long scales) the same criteria were tested in both studies. This reduced the number of tables and avoided redundancies.

Table 2. Most of the descriptions are inconsistent with the results shown in Table 2. For example, also Agreeableness is negative.

Special thanks for your attention and this note. There was indeed an error regarding the direction of a correlation. We checked and re-formulated the whole paragraph (p. 10, lines 389-397).

The two short scales correlated differently with the external variables in many cases. Avoidance showed the highest significant correlations with relationship satisfaction, erotic love, and conscientiousness (negative) and with schizoid personality disorder and emotional abuse and neglect (positive). The effect sizes were moderate. Anxiety showed the highest significant correlations with possessive love, stress, general anxiety, somatization, neuroticism, and paranoid, histrionic, narcissistic, dependent, and compulsive personality disorder (all positive). The effect sizes were moderate to strong. The anxiety short scale therefore showed higher and more statistically significant correlations than the avoidance short scale, a pattern of results that was also found in the long scales.

It is also essential to clarify why most results are insignificant!

In our view, it is positive that not all correlations became significant and, in particular, that some of the scales correlated with only one of the two ECR short scales. This proves the discriminant validity of the two scales. This aspect is discussed on p. 13, lines 524-536.

Reviewer 2 Report

Comments and Suggestions for Authors The present contribution is rather a nice 3-experiment study, containing some original data. Far from representing the mere translation in German language of a rather widely exploited psychological Scale, it may inform the vast audience of BS on several important issues. Therefore it is necessary to better explain some parts, to render the paper more understandable for the average behavioural scientist.  

Specific comments :

Page 1, lines 15-16 

the item pool of the original version based on ant colony optimization (ACO). Data from three samples collected at a university, an online portal, 

The fact that a human psychological scales is derived by ant collective behavioural patterns absolutely necessitates of one or two paragraphs narrating such an original methodological history.

Page 2, lines 60-62 

 Furthermore, significant correlations with measures 60 of personality and psychopathology could be shown. The ECR has been translated into 17 languages.  

Personality issues are very trendy in contemporary human (and animal) psychology. A much more detailed review of such a correlation, both at the "typical" and pathological level seems necessary.

Lines 82-82 

Moreover, participants can sometimes not be expected to complete a very long survey, for example if they do not receive payment or course credit or if they are mentally ill. We did not consider using the short forms of 

To include mentally ill among participants to a psychological study indicates definitely a strange design.

Page 7, lines 276-277 

For this purpose, we re-analyzed data from former studies that used the ECR-G-36 [18,24,25,26]. In these studies, the ECR 

It is not clear, at least at a first glance, how those data were selected and reused.

Page 10, Table 

Histrionic PD .21* .18 .48*** .45*** 

Narcissistic PD .19 .19* .43*** .44*** 

For those two rather abused, among nonspecialist readers, terms a clear definition is needed, eg in the Methods section.

Page 13, lines 434-437 

Satisfaction with Sexuality (SWS). Satisfaction with sexuality in the romantic relationship was assessed with the following item formulated by Rohmann and Bierhoff [44]: “How satisfied are you with your sexual relationship with your partner?” The response format is the same as for the RAS

Again, for nonspecialist readers, and also more in general, such a simplistic way of assessment do need a better clarification.  What about eg for not strictly monogamous, yet "romantic", couples?

Author Response

Response to Reviewer 2

Thank you for reviewing our manuscript and the constructive feedback. Please find the detailed responses below. Revised and newly formulated passages are quoted here and shown in red.

The present contribution is rather a nice 3-experiment study, containing some original data. Far from representing the mere translation in German language of a rather widely exploited psychological Scale, it may inform the vast audience of BS on several important issues. Therefore it is necessary to better explain some parts, to render the paper more understandable for the average behavioural scientist.

We especially appreciate that you see the contribution of our work not only in the development of a German short version of the ECR, but also acknowledge substantive contributions to attachment research.

Specific comments:

Page 1, lines 15-16 

the item pool of the original version based on ant colony optimization (ACO). Data from three samples collected at a university, an online portal, 

The fact that a human psychological scale is derived by ant collective behavioural patterns absolutely necessitates of one or two paragraphs narrating such an original methodological history.

We have added some details including a new reference in the two paragraphs describing this method to make it understandable that this algorithm was derived and named by observing ants and is best understood as a probabilistic mathematical problem solving method (pp. 4,5, lines 192-207).

The ACO provides a probabilistic algorithm for this purpose, which was reported and mathematically described by Feynman [26], from which its denotation is derived. This class of algorithms is based on an iterative and combinatorial process that imitates ants searching for food. For this purpose, ants aim to find the shortest path marked by a chemical trace of pheromones. Because these pheromones are constantly evaporating, shorter paths have a higher concentration of them and are therefore more attractive to following ants. The shortest path in the context of determining an item selection can be operationalized as an optimal quality criterium that must be determined mathematically.

The ACO starts with the selection of a random sample based on all suitable items. Then, predefined quality criteria of interest (e.g., model fit and reliability) of this sample are determined, and subsequently a large number of different randomly selected samples are compared. The algorithm leaves the best at the top to identify an optimized solution. Considering the fact that this probabilistic approach cannot determine a theoretically existing optimal solution in every case, a series of repetitions of this procedure is recommended [25].

Page 2, lines 60-62

Furthermore, significant correlations with measures 60 of personality and psychopathology could be shown. The ECR has been translated into 17 languages.

Personality issues are very trendy in contemporary human (and animal) psychology. A much more detailed review of such a correlation, both at the "typical" and pathological level seems necessary.

We have expanded the paragraph on the external validity of the ECR (p. 2, lines 61-75). Studies and reviews on the correlations with romantic relationships, normal and pathological personality and mental disorders have now been included.

The ECR has been proven valid in hundreds of studies [3]. The two scales correlated as expected with other measures of romantic relationships. A variety of studies with different research designs (e.g., cross-sectional and longitudinal studies, diary studies, experiments) found that low scores on avoidance and anxiety were associated with a positive view of the relationship and higher satisfaction with the relationship and sexuality. [5]. Furthermore, significant correlations with measures of personality and psychopathology could be shown. For example, the ECR scales were found to correlate with the Big Five of personality, with the strongest correlation observed between anxiety and neuroticism [6]. Such associations have also been demonstrated for pathological personality traits in psychiatric patients. Here, the highest correlations were found between anxiety and negative affectivity and between avoidance and detachment [7]. Finally, the ECR scales turned out to correlate with measures of mental disorders. Many of the studies in this area have focused on depression, anxiety, or borderline personality disorder, with all three of these mental disorders showing positive associations with attachment avoidance and anxiety [3,8].

Lines 82-82

Moreover, participants can sometimes not be expected to complete a very long survey, for example if they do not receive payment or course credit or if they are mentally ill. We did not consider using the short forms of 

To include mentally ill among participants to a psychological study indicates definitely a strange design.

In the clinical samples of our study, there were indeed participants with mental illness. All respective studies followed the guidelines of good clinical practice and were approved by the local ethics committee. We therefore believe that the inclusion of mentally ill participants was low risk for them. We have reformulated the relevant sentence (p. 2, lines 95-97) and are now concentrating on the probably limited resilience of these participants.

Furthermore, sometimes participants cannot be expected to complete a lengthy survey, for example, if they do not receive payment or course credit or if they have limited resilience due to mental illness.

Page 7, lines 276-277 

For this purpose, we re-analyzed data from former studies that used the ECR-G-36 [18,24,25,26]. In these studies, the ECR 

It is not clear, at least at a first glance, how those data were selected and reused.

Since these data were not newly collected but were collected for previous studies and reanalyzed in the present study, we have provided only a brief note on the data collection processes under the heading “Participants and settings.” Further information can be found in the publications cited. We didn't want to make the text too long and therefore decided on this approach in the current study. Please let us know if you think this is inadequate and it is necessary to provide detailed information also in the current study.

Page 10, Table 

Histrionic PD .21* .18 .48*** .45*** 

Narcissistic PD .19 .19* .43*** .44*** 

For those two rather abused, among nonspecialist readers, terms a clear definition is needed, eg in the Methods section.

Here we have decided not to present these concepts in more detail, but rather to refer to the relevant publications. This was done again in order not to make the text too long. It would have required longer passages of text to describe the 10 personality disorders and other concepts such as the Big Five in more detail.

Page 13, lines 434-437 

Satisfaction with Sexuality (SWS). Satisfaction with sexuality in the romantic relationship was assessed with the following item formulated by Rohmann and Bierhoff [44]: “How satisfied are you with your sexual relationship with your partner?” The response format is the same as for the RAS

Again, for nonspecialist readers, and also more in general, such a simplistic way of assessment do need a better clarification.  What about eg for not strictly monogamous, yet "romantic", couples?

We agree that the questionnaires assessing romantic relationships and sexuality are more aimed at people in a monogamous relationship. We raised this point under “Limitations” in the discussion, noting that participants in open relationships with more than one partner are likely to find it difficult to complete these questionnaires.

Furthermore, the scales for the assessment of the romantic relationship and sexuality may have been difficult to answer for participants who were in an open relationship and felt connected to more than one partner.

Reviewer 3 Report

Comments and Suggestions for Authors

The authors of this article developed a German version of the Experiences in Close Relationships scale. They adequately justified the study and described the three different studies they conducted to achieve their goal and to establish credible psychometrics for their study. 

Their approach and analyses were sound. In particular, their approach to Ant Colony Optimization was sound and intriguing.

The discussion of the results of their studies was appropriate.

The tables included in the manuscript are hard to read with odd and unpredictable spacing and centering. This will limit readers' abilities to interpret the work and findings. 

Author Response

Response to Reviewer 3

Thank you for reviewing our manuscript and the constructive feedback. Please find the detailed responses below.

The authors of this article developed a German version of the Experiences in Close Relationships scale. They adequately justified the study and described the three different studies they conducted to achieve their goal and to establish credible psychometrics for their study. 

Their approach and analyses were sound. In particular, their approach to Ant Colony Optimization was sound and intriguing.

The discussion of the results of their studies was appropriate.

Thank you very much for this positive assessment. We are pleased that you appreciate our contribution.

The tables included in the manuscript are hard to read with odd and unpredictable spacing and centering. This will limit readers' abilities to interpret the work and findings. 

The layout of the tables was indeed not consistent, and the tables were therefore confusing. We have given all tables a uniform layout and also prevented a table from being split into two pages. We hope that the tables are now more readable.